# Hardened Steels and Their Machining

**Karel Osička \*** and **Josef Chladil**

Faculty of Mechanical Engineering, Brno University of Technology, Technická 2996/2, 61602 Brno, Czech Republic; chladil@fme.vutbr.cz
\* Correspondence: osicka@fme.vutbr.cz

**Abstract:** This article discusses the issue of hardened steel machining. Many components in the engineering industry use hardening as the final heat treatment. These components usually occupy a significant position in a given assembly unit. They guarantee the correct operation of the entire technical equipment in total cooperation with other components. The quality of these components depends on the integrity of their surface. The production of these parts is usually carried out by traditional technological procedures. Clearly, an example of such a technology is grinding. However, this article discusses the application of other finishing technologies using a tool material made of cubic boron nitride (CBN). The technology used is finishing turning with subsequent tumbling technology. The subject of the evaluation is the integrity of the surface. In this part of the experiments, there are mainly individual parameters of surface roughness. Compared components are bearing rings, in this case the inner surface of the housing ring.

**Keywords:** finishing; turning; hardened steel; cubic boron nitride; bearing ring

## 1. Introduction

The machining of hardened steels is the subject of research at the Institute of Manufacturing Technology, Faculty of Mechanical Engineering, Brno University of Technology. The subject of the research is the search for optimal cutting conditions, the assessment of the influence of tool overhang and the assessment of the stability of the cutting process for the selected machining method. The results were separately published in the following articles: "Machining of hardened bearing steels" [1], "Contribution to turning hardened steel" [2], "Influence of cutting tool overhangs at machining of hardened steels" [3], "High-speed cutting of bearing rings from material 100Cr6" [4], Analysis of selected aspects of turned bearing rings regarding required work piece quality" [5] and "Tension of the surface layer in machining hardened steels" [6]. The article does not deal with the shape of the resulting chips or force ratios. These matters are already described in detail in "Turning of Hardened Steels" [7] by Prof. Malkin. Basic data on the machining of hardened steels, such as the effect of temperature on wear, force loads when using both sintered carbides and cubic boron nitride (CBN)material, are already known "Wear and tool life of tungsten carbide, PCBN and PCD cutting tools" [8], "Cutting Temperature and tool wear of hard turning hardened bearing steel" [9], "Influence of Different Grades of CBN Inserts on Cutting Force and Surface Roughness of AISI H13 Die Tool Steel during Hard Turning Operation" [10], "An overview on economic machining of hardened steels by hard turning and its process variables." [11], "Machining of hardened steel—Experimental investigations, performance modeling and cooling techniques." [12] and "A review of turning of hard steels used in bearing and automotive applications." [13]. The application research to date has been focused mainly on machining the outer surface, i.e., the functional surface of the shaft ring. This article deals mainly with surface roughness parameters after finishing by internal turning with a cubic boron nitride tool. Part of the samples are further treated with tumbling technology. The machining results were statistically evaluated with a focus on the stability of the machining process. The application of CBN material to hardened steel

machining is a known fact, but the specific exact machining conditions for each type of steel are not commonly known. The purpose of the experiment is to assess the technology of production used for two samples made of hardened steel. Sample No.1 was worked by turning the cutting edge of the CBN material and then tumbling. Sample No.2 was produced by only turning the cutting edge of the CBN material. The finishing technology used so far after hardening has been grinding. This technology is tested and guarantees accuracy of dimensions and integrity of the surface, including pressure stress in the surface layer. The main objective of the experiment is therefore to find a suitable combination of cutting conditions for hard turning and subsequent tumbling. At the same time, this combination must not significantly affect the durability of the tool.

## 2. Materials and Methods

The experiment is focused on the production of the functional surface of the "Housing Ring" component, which is the inner diameter. The realized production process includes:

- Turning hardened materials with a tool from CBN
- Tumbling technology of the surface of hardened components
- Evaluation of the surface integrity of the functional surface.

The experiment was carried out using the machinery and laboratory equipment at the following workplaces:

- Faculty of Mechanical Engineering, Brno University of Technology (turning center SP280 SY/Sinumerik 840D, rough gauge Taylor Hobson Surtronic S 100)
- Flidr plast. (tumbling equipment TVS 1100 × 460, rough gauge Mitutoyo SJ-201)
- Faculty of Nuclear and Physical Engineering of the Czech Technical University in Prague (X-ray analysis)

Material of bearing ring according to International Organization for Standardization is "100 Cr6". Material composition is in Table 1. The hardness of the samples is 62 to 64 HRC.

**Table 1.** Material composition "100 Cr6" (%).

| Material | C | Si | Mn | Cr | Mo | P | S |
|---|---|---|---|---|---|---|---|
| 100Cr6 | 0.93–1.05 | 0.15–0.35 | 0.25–0.45 | 1.35–1.60 | max.0.1 | 0.025 | 0.015 |

The 100Cr6 steel is hot-working and suitable for direct hardening. This steel in a soft annealed state is well machined. The optimal diameter or thickness of the seasoning is about 20 mm. Corrosion resistance is normal. Steel is suitable for components with a very hard, wear-resistant surface. This steel is also suitable for the production of balls up to 25 mm in diameter, rollers and cones up to 18 mm in diameter and rings of rolling bearings of smaller size. Basic data for the processing of semi-finished product:

- forging 750 to 1100 °C
- normalization annealing 860 to 890 °C
- soft annealing 720 to 760 °C
- hardening into water 790 to 820 °C
- hardening into oil 820 to 850 °C
- tempering 150 to 220 °C

  Indicative sample dimensions:

- Inner diameter 145 mm
- Length 102 mm

The cubic boron nitride insert CNGA120412S-01525L1WZB, CBN160C by Seco Tools AB (catalogue p. 480) is used for finishing the inner diameter of the bearing ring. The corresponding knife holder from Seco Tools AB type S25S-MCLNR12 (catalogue p. 374) is shown in Figure 1 below in the working area of the SP280SY. The CBN insert is also shown here, as are the magnification details of cutting edge No.1 and cutting edge No.2. In both

cases, you can see a slight wear on the head of the tool. In both cases, the cutting edge is retained.

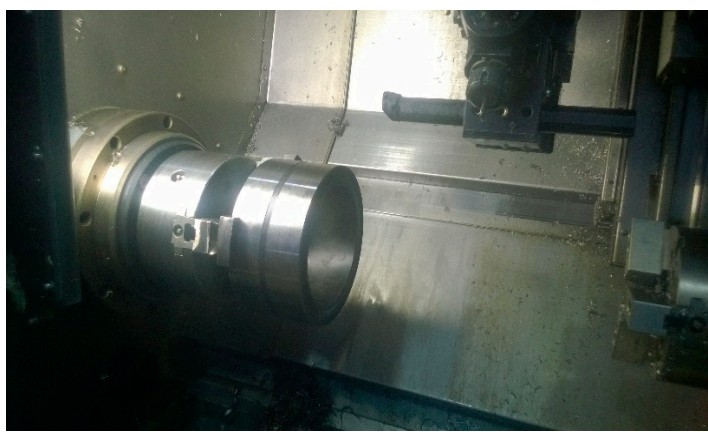
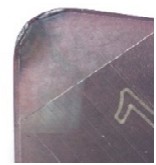
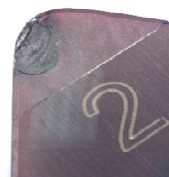
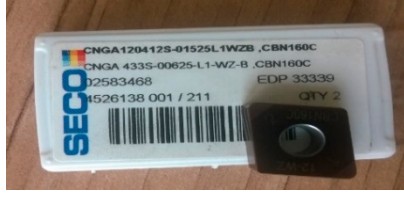

**Figure 1.** The machine SP280 SY workspace and details of cutting edge No.1 and No.2 inserts from cubic boron nitride (CBN).

Cutting conditions and machining times:

| | |
|---|---|
| Blade width $a_p$ (mm) | 0.2 |
| Cutting speed $v_c$ (m·min$^{-1}$) | 155 |
| Feed f (mm) | 0.05–0.07 |
| Outer diameter $D_e$ (mm) | 212.6 |
| Outer length $l_e$ (mm) | 51 |
| Inner diameter $D_i$ (mm) | 162.6 |
| Inner length $l_i$ (mm) | 102 |

The machining process of each sample is given in the following tables. Removal of surface layer from the inside, roughing time $t_h$ is given in Table 2.

**Table 2.** Roughing time $t_h$.

| Conditions | Feed f (mm) | Cutting Speed $v_c$ (m/min) | Inner Diameter Di (mm) | Inner Length $l_i$ (mm) | Cutting Time $t_h$. (min) | Note. |
|---|---|---|---|---|---|---|
| | 0.15 | 155 | 162.6 | 102 | 2.24 | |

The cutting tool: CNGA120412S-01525L1WZB, CBN160C, cutting edge No.2. The above cutting conditions have previously been verified in the laboratory of mechanical machining at the Institute of Engineering Technology of the Faculty of Mechanical Engineering, Brno University of Technology, and substantially refine the recommended values of the tool manufacturer. These recommended values have a range that is too large and do not affect the resulting state of surface tension. The tensile state of surface tension arises at cutting speeds in the range of 125–145 m·min$^{-1}$. These speeds are suitable for less tool wear. The pressure state of surface tension arises at higher cutting speeds in the range of 155–195 m·min$^{-1}$. These speeds, mainly in the range of 175–195 m·min$^{-1}$, are the cause of greater tool wear. The selected cutting speed of 150 m·min$^{-1}$ is a compromise between the requirement for even more acceptable tool wear and pressure surface tension; the size of the pressure tension is limiting at a cutting speed of 150 m·min$^{-1}$ and can be improved by tumbling technology.

Full-length finish from the inside, finishing time $t_d$ is given in Table 3.

The cutting tool: CNGA120412S - 01525L1WZB, CBN160C, cutting edge No.1.

For sample No.1, a tumbling operation was subsequently performed in cooperation with the company Flídr under the following conditions:

- Tumbling machine: TVS 1100 × 460
- Tumbling balls RST 8 G

- Process fluid Compound FC 122 A (concentration 3%)
- Hot air drying
- Time 10 h in a paste without flow and 4 h, rinsing 50 l/hour.

**Table 3.** Finishing time $t_d$.

| All Samples | Feed f (mm) | Cutting Speed $v_c$ (m/min) | Inner Diameter Di (mm) | Inner Length $l_i$ (mm) | Time $t_d$. (min) | Note. |
|---|---|---|---|---|---|---|
| 1 | 0.05 | 155 | 162.6 | 102 | 6.72 | |
| 2 | 0.07 | 155 | 162.6 | 102 | 4.80 | |

## 3. Results

A complete evaluation was carried out on components 1 and 2. Evaluation of the sample surfaces from the inside was carried out by measuring the surface roughness parameters on the Taylor Hobson Surtronic s 100 as shown in Figure 2. The quantities according to ISO4287 and the carrier share according to ISO13565 were selected from the range of possible measured values. Measurements were made at 6 different locations on each sample, allowing the 16% rule to be applied, where a given surface can be accepted if not more than 16% of all the measured parameter values exceed the value indicated in the drawing or production documentation. Practically, this means that no more than one of the first six measured values exceeds the specified value.

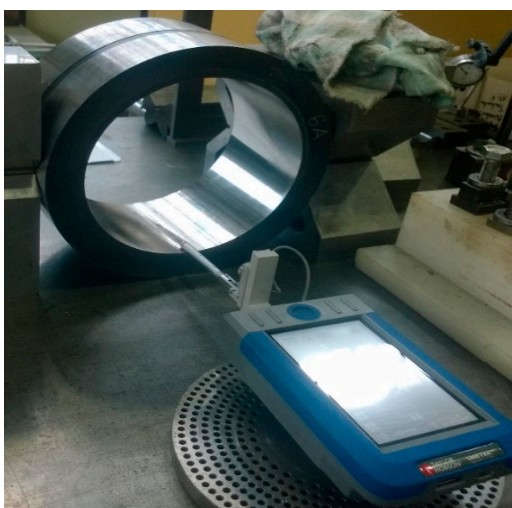

**Figure 2.** Measurements on the touch device Taylor Hobson Surtronic S 100.

X-ray analysis was carried out to a limited extent on the surface of the internal diameter for samples No.1 and No.2 at the Department of Solid State Engineering of the Faculty of Nuclear and Physical Engineering of the Czech Technical University in Prague.

The diffusing measurement was carried out on the θ-θ goniometer X ′Pert PRO MPD by PANalytical. Two different experimental arrangements, i.e., semi-focused Bragg–Brentan and tangent bundle were used to determine macroscopic residual stresses, microdeformations and crystalline size [14]. The multi hkl and sin 2ψ method was used to calculate residual stresses [14]. Rietveld's refinement methods were used to determine microstructure parameters. The inner surface of the component does not allow perpendicular adjustment of the X-ray head, as shown in Figure 3.

### 3.1. Post-Machining Evaluation

Evaluation of the inner surface after turning is performed on each part by 6 measurements. The Tables 4–6 below show statistical variables such as standard deviation, mean, median and mode. The display of the device is shown in Figures 4–6 with a chart for one of the 6 measurements, in which the most frequently measured value is found.

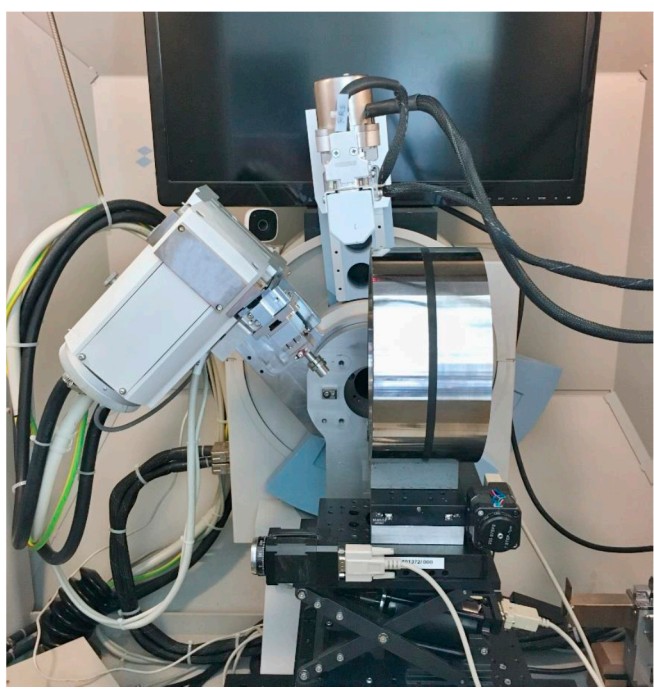

**Figure 3.** Sample when analyzing the inner surface.

Note: Meaning of abbreviations

According to ISO4287:

| | |
|---|---|
| Ra | Average arithmetic deviation of the profile |
| Rz | Maximum profile height |
| Rmr | Mutual material ratio for (Mr = 50%, an offset Rδc = 0.1 μm) |
| Rv | Maximum depth of profile tips |
| Rp | Maximum height of profile tips. |

According to ISO13565:

| | |
|---|---|
| Rk | Average arithmetic profile aversion |
| Rpk | Maximum profile height |
| Rvk | Mutual material ratio for (Mr = 50%, an offset Rδc = 0.1 μm) |
| Mr1 | Maximum depth of profile tips |
| Mr2 | Maximum height of profile tips |

Note: "x" in Tables 4–6 indicates that the appropriate amount cannot be displayed for the set of values.

**Table 4.** Component 1.

| Measurement/Quantities | Ra (μm) | Rz (μm) | Rmr (%) | Rk (μm) | Rpk (μm) | Rvk (μm) | Mr1 (%) | Mr2 (%) | Rv (μm) | Rp (μm) |
|---|---|---|---|---|---|---|---|---|---|---|
| 1 | 1.21 | 5.00 | 49.80 | 3.70 | 1.80 | 0.30 | 19.10 | 98.40 | 2.00 | 3.00 |
| 2 | 0.24 | 1.10 | 47.90 | 0.70 | 0.60 | 0.40 | 13.80 | 82.50 | 0.50 | 0.50 |
| 3 | 0.18 | 1.00 | 48.90 | 0.70 | 0.90 | 0.30 | 18.20 | 93.20 | 0.40 | 0.60 |
| 4 | 0.16 | 1.00 | 48.30 | 0.50 | 0.70 | 0.30 | 16.40 | 93.50 | 0.40 | 0.60 |
| 5 | 0.24 | 1.20 | 45.80 | 0.50 | 0.70 | 0.20 | 31.80 | 95.30 | 0.40 | 0.80 |
| 6 | 0.27 | 1.20 | 51.30 | 0.80 | 0.90 | 0.20 | 29.70 | 97.20 | 0.50 | 0.70 |
| Deviation | 0.41 | 1.59 | 1.86 | 1.25 | 0.44 | 0.08 | 7.42 | 5.69 | 0.64 | 0.97 |
| Average | 0.38 | 1.75 | 48.67 | 1.15 | 0.93 | 0.28 | 21.50 | 93.35 | 0.70 | 1.03 |
| Median | 0.24 | 1.15 | 48.60 | 0.70 | 0.80 | 0.30 | 18.65 | 94.40 | 0.45 | 0.65 |
| Modus | 0.24 | 1.00 | x | 0.70 | 0.90 | 0.30 | x | x | 0.40 | 0.60 |

Partial evaluation of the average arithmetic deviation of the profile Ra.

- For component No.1, the extreme surface roughness value for the 1st measurement, which greatly affects the diameter and standard deviation from statistical quantities. Other statistical quantities of median and modus indicate a relatively better Ra value on most surfaces. The condition is caused by accidental oscillation of the instrument during captivity into the material.

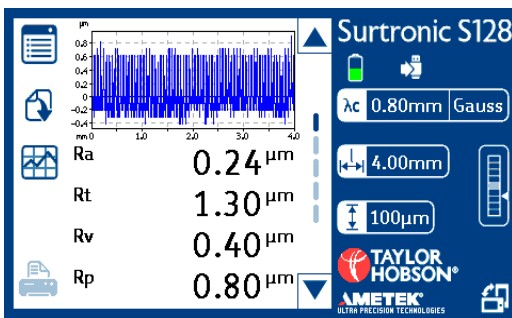

**Figure 4.** Chart for mode Ra 0.24.

**Table 5.** Component 2.

| Measurement/Quantities | Ra (μm) | Rz (μm) | Rmr (%) | Rk (μm) | Rpk (μm) | Rvk (μm) | Mr1 (%) | Mr2 (%) | Rv (μm) | Rp (μm) |
|---|---|---|---|---|---|---|---|---|---|---|
| 1 | 0.23 | 1.60 | 52.80 | 1.50 | 1.50 | 0.30 | 11.00 | 97.40 | 0.60 | 0.90 |
| 2 | 0.25 | 1.60 | 55.90 | 1.50 | 1.20 | 0.40 | 7.50 | 93.90 | 0.70 | 0.90 |
| 3 | 0.29 | 2.00 | 56.30 | 2.20 | 1.00 | 2.00 | 7.80 | 76.80 | 1.10 | 1.00 |
| 4 | 0.27 | 1.60 | 54.90 | 1.00 | 1.40 | 0.40 | 23.90 | 95.30 | 0.70 | 0.90 |
| 5 | 0.29 | 1.60 | 54.90 | 1.80 | 1.40 | 0.40 | 7.80 | 97.20 | 0.70 | 0.90 |
| 6 | 0.25 | 1.60 | 52.50 | 1.20 | 1.40 | 0.40 | 14.50 | 90.60 | 0.60 | 1.00 |
| Deviation | 0.02 | 0.16 | 1.57 | 0.43 | 0.18 | 0.66 | 6.39 | 7.79 | 0.19 | 0.05 |
| Average | 0.26 | 1.67 | 54.55 | 1.53 | 1.32 | 0.65 | 12.08 | 91.87 | 0.73 | 0.93 |
| Median | 0.26 | 1.60 | 54.90 | 1.50 | 1.40 | 0.40 | 9.40 | 94.60 | 0.70 | 0.90 |
| Modus | 0.25 | 1.60 | 54.90 | 1.50 | 1.40 | 0.40 | 7.80 | x | 0.70 | 0.90 |

Partial evaluation of the average arithmetic deviation of the profile Ra.

- For component No.2, the median diameter and modus stability of the machining process can be seen at almost the same statistical values. The standard deviation is very small.

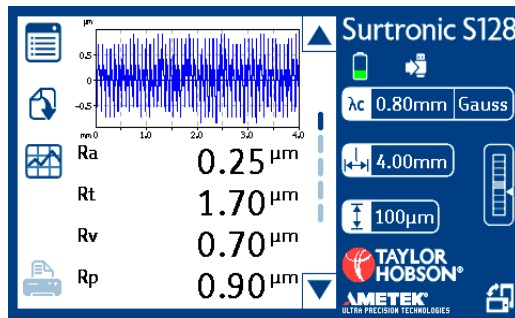

**Figure 5.** Chart for mode Ra 0.25.

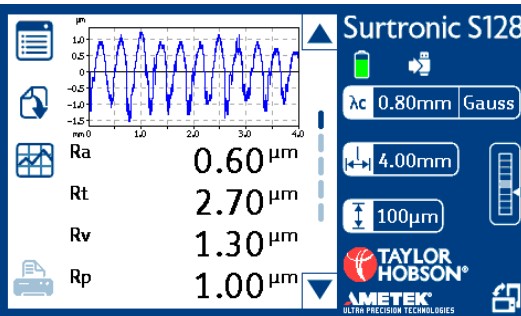

**Figure 6.** Chart for mode Ra 0.6.

**Table 6.** Sample No.1 after tumbling.

| Measurement/Quantities | Ra (µm) | Rz (µm) | Rmr (%) | Rk (µm) | Rpk (µm) | Rvk (µm) | Mr1 (%) | Mr2 (%) | Rv (µm) | Rp (µm) |
|---|---|---|---|---|---|---|---|---|---|---|
| 1 | 0.62 | 2.60 | 48.50 | 2.50 | 1.90 | 1.00 | 13.10 | 87.50 | 1.40 | 1.30 |
| 2 | 0.60 | 2.40 | 49.80 | 2.50 | 1.30 | 0.80 | 11.00 | 88.90 | 1.30 | 1.10 |
| 3 | 0.60 | 2.40 | 51.80 | 2.30 | 1.50 | 0.60 | 19.80 | 87.40 | 1.30 | 1.00 |
| 4 | 0.07 | 0.60 | 50.90 | 1.00 | 0.20 | 0.40 | 7.80 | 99.00 | 0.30 | 0.30 |
| 5 | 0.15 | 0.80 | 48.50 | 1.00 | 0.00 | 2.80 | 0.10 | 49.80 | 0.30 | 0.50 |
| 6 | 0.16 | 0.90 | 51.70 | 0.80 | 0.20 | 5.90 | 6.30 | 58.70 | 0.30 | 0.50 |
| Deviation | 0.26 | 0.94 | 1.50 | 0.83 | 0.81 | 2.13 | 6.67 | 19.52 | 0.57 | 0.40 |
| Average | 0.37 | 1.62 | 50.20 | 1.68 | 0.85 | 1.92 | 9.68 | 78.55 | 0.82 | 0.78 |
| Median | 0.38 | 1.65 | 50.35 | 1.65 | 0.75 | 0.90 | 9.40 | 87.45 | 0.80 | 0.75 |
| Modus | 0.60 | 2.40 | 48.50 | 2.50 | 0.20 | x | x | x | 0.30 | 0.50 |

### 3.2. Evaluation after Tumbling Technology

3.2.1. Evaluation of Surface Quality of Tumbling at Firm Flídr.

The post-tumbling evaluation was carried out on the Mitutoyo SJ-201 rougheameter with the result in sample No.1. of Ra—0.05 (0.06). The measurement was made in an area outside the tool oscillation zone.

3.2.2. Evaluation after Tumbling on the Rough Gauge Taylor Hobson Surtronic S 100

The above measurement in Section 3.1 was subsequently compared with the measurement of the roughness of the surface of the component samples on the touch device of the Institute of Engineering Technology.

Partial evaluation of the average arithmetic deviation of the profile Ra:

- Component No.1 is the extreme negative surface roughness value for three measurements and a very good value for measurements for No.4. The condition is caused by the input state after machining (see the rating for Table 5, where the extreme roughness value for measurement No.1 was). The condition after tumbling improved greatly, however. On that part of the surface where a similar Ra value (0.05 and 0.06) was measured on both instruments, the tumbling technology was no longer fundamentally reflected. Statistical quantities are influenced by a large variance of values.

### 3.3. X-ray Analysis

The X-ray analysis was carried out to a limited extent on the surface of the internal diameter for samples No.1 (remedied) and No.2 (only machined) at the Department of Solid-State Engineering of the Faculty of Nuclear and Physical Engineering of the Czech Technical University. The result is influenced by the oblique adjustment of the X-ray head, and the measurement is only in the axial direction, as is shown in Figure 3. Values of macroscopic residual stresses are designed in Table 7.

**Table 7.** Values of macroscopic residual stresses $\sigma \pm \Delta\sigma$.

| Method | Part No.1 (MPa) | Part No.2 (MPa) |
|---|---|---|
| Co – {310} | $-1070 \pm 48$ | $-637 \pm 36$ |
| Cr – {211} | $-1203 \pm 38$ | $-563 \pm 58$ |

The values in the table show twice the value of the macroscopic compressive residual stress of the tumbling component in both X-ray analysis methods.

## 4. Discussion

The article deals only with a partial part of the overall development and evaluation of the integrity of a surface after turning hardened steels. The ultimate purpose is to replace the finishing technology with CBN grinding technology in a comparable way. This comparable technology must ensure not only the roughness parameters but, above all, the pressure state of the tension in the surface. This surface condition is important for components as highly stressed as bearing rings. Another aspect that must be assessed on the surface of the bearing rings are tribological properties, namely the ability of the surface to maintain the lubricating mass. This tribological property of the surface is guaranteed by grinding technology (cross-cutting), which leaves fine grooves along the grinding grains. These fine grooves can subsequently keep trace amounts of the lubricating medium on the surface. The cutting edge from CBN does not leave such micro-grooves, and therefore, it is necessary to solve the issue of tribology in parallel. Another risk is the verification of the state of tension in the surface layer in the serial production of bearing rings. Here a simple comparison method is used to evaluate the state of tension (so-called Barkhausen noise). This method can compare mass-produced pieces with standard material. This standard must be verified by X-ray analysis. X-ray analysis is a time-consuming and costly method that cannot be used in serial production.

## 5. Conclusions

The individual partial results of the experiment confirm the following:

- Better surface quality is achieved at lower feed rates of 0.05–0.07 mm.
- Tumbling will significantly affect the surface quality only the in case of worse input values Ra.
- The machined surface could be accepted sample 1 if we applied the 16% rule and excluded from the evaluation the marginal area where the vibrations occurred.
- Tumbling practically doubles the compressive macroscopic residual stress, which is a very good result in terms of surface integrity.
- The requirement for functional surfaces of bearing ring is Ra < 0.2, which would be met only at part No.1 assuming the elimination of vibrations using another tool.

The effect of tumbling technology on the average arithmetic of the Ra profile is difficult to demonstrate, especially on surfaces where the default Ra value is good, i.e., less than 0.2. The effect of tumbling on the doubling of the residual pressure voltage is clearly demonstrated on one evaluated sample. The next direction of development will be focused on the bearing shaft ring component, where conditions are significantly better both in terms of machining conditions of the outer surface and in terms of evaluation especially for X-ray analysis, where we can make measurements in axial and radial directions with perpendicular adjustment of the X-ray head. The results of X-ray analysis of the outer surface (shaft ring) can then be compared and possibly generalized with the results of the inner surface (housing ring). The main objective of the presented experiment is to find the optimal parameters of hard turning technology and subsequent tumbling. These parameters must also be economically advantageous. It will then be possible to replace the finishing technology of grinding even for such stressed bearing rings.

**Author Contributions:** Conceptualization, K.O.; methodology, J.C. All authors have read and agreed to the published version of the manuscript.

**Funding:** This research was funded by the project FV 40225, program TRIO, Ministry of Industry and Trade of the Czech Republic.

**Data Availability Statement:** Not applicable.

**Acknowledgments:** Samples of semi-finished hardened material were provided by firm ZKL, tumbling technology by firm Flidr and X-ray diffraction was carried out at the Faculty of Nuclear and Physical Engineering of the Czech Technical University.

**Conflicts of Interest:** The funders had no role in the design of the study; in the collection, analyses, or interpretation of data; in the writing of the manuscript, or in the decision to publish the results. The authors declare no conflict of interest.

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
