# Peer review of "Hardened Steels and Their Machining"

_crystals, doi:10.3390/cryst11020182_

Round 1

Reviewer 1 Report

The manuscript focuses on machining hardened steel, precisely on the selection of parameters in a way that allows obtaining the best roughness parameters. After correction article is better but still requires work. Compared to the previous version of the manuscript, the corrections made by the authors are too minor. I have the following comments:

  • Authors should expand the introduction and the literature review
  • Authors should indicate what is a novelty
  • The discussion and summary section should be expanded.
  • References should be enriched with publications not belonging to the authors.
  • I still do not understand the interest of the tests. The authors must justify the interest

Author Response

Hello Mr. Reviewer,

Thank you for the inspiring comments on our article.

 I extended:

  • the introduction,
  • the literature review
  • the discussion
  • the conclusion

The main objective of the presented experiment is to find the optimal parameters of hard turning technology and subsequent tumbling. These parameters must also be economically advantageous. It will then be possible to replace the finishing technology of grinding even for such stressed bearing rings.

Regards

Karel Osička

Reviewer 2 Report

Totally the present article is well-established and the subject is interesting, but some minor revisions should be considered.

  1. It is suggested to present the structure of the article at the end of the introduction.
  2. The literature review is not enough. There some articles, which can be added to a literature review. For example:
    - Kumar, Pardeep, et al. "Influence of different grades of CBN inserts on cutting force and surface roughness of AISI H13 die tool steel during hard turning operation." Materials 12.1 (2019): 177.

  3. A more detailed description of the workpiece material is recommended. In terms of where it is applied, machinability...
  4. The authors can explain in more detail how they define the cutting parameters. 
  5. If possible, the authors can insert an X-ray image.

Author Response

Hello Mr. Reviewer,

Thank you for the inspiring comments on our article.

I made subsequent adjustments:

  • the structure of the article
  • the extension of literature
  • a detailed description of the material
  • an explanation of the cutting conditions

As for X-ray scans, these were not carried out. Data from diff fractional graphs were used to determine residual stresses.

Regards

Karel Osička

Round 2

Reviewer 1 Report

After updates and corrections, the manuscript is much clearer, revised explanations make it easier to understand. The authors responded to all the comments and made the necessary corrections. I recommend changing commas to periods in tables and text in floating-point numbers.

This manuscript is a resubmission of an earlier submission. The following is a list of the peer review reports and author responses from that submission.

Round 1

Reviewer 1 Report

From mentioned article is not obviously aim, with wich was measurement doing. Some informations repeat themselves, e.g. used cutting tool and insert. Some terms woud be further explain, e.g. edge No. 1 and No. 2, difference bettwen sample No. 1 and No. 2 (component No. 1 and No. 2)? I miss mention of foreign publications in references which deal with followed issues. Tumbling technology show totally different results with propably difficult repeatability. On the side 4 - The display of device is shown in Figure 4-6, not in Figure 5-7 as is mentioned in text. Roughness units in tables 4-6 are not exact.

Reviewer 2 Report

The manuscript focuses on machining hardened steel, precisely on the selection of parameters in a way that allows to obtain best roughness parameters. The article requires many improvements. I have the impression that the authors have carried out an experiment that was not entirely thought out and described it. In view of this manuscript, I have the following comments:

  • Authors should expand the introduction and the literature review
  • Authors should indicate what is novelty
  • The authors should indicate what is new compared to other studies. The issue of hardening steel machining is quite well known
  • The discussion and summary section should be expanded.
  • How the machining parameters were selected and what was the plan of the experiment. What the authors wanted to prove with their research?
  • References should be enriched with publications not belonging to the authors.
  • I do not understand the interest of the tests. The authors must justify the interest.
  • The text must be reread to correct errors and repetitions. English needs to be improved. More generally, the writing needs to be revised to reach the level of a high-level scientific article.

Reviewer 3 Report

The authors investigated surface roughness of the inner surface of a housing ring by changing a tool material and employing tumbling technology as an additional surface treatment. Even though the authors did lots of works, the manuscript should be improved and re-organized. Most of all, there is no scientific information which has to be provided in the manuscript. For example, in the conclusion section, the authors addressed only future work instead of new findings in your work. The manuscript looks like a technical report (only deliver data you obtained without discussion). Please re-construct your data to get valuable information and then, make conclusions based on the discussion on your data.